# Effects of precise cardio sounds on the success rate of phonocardiography

**Youngsin Kim[1], Mihyung Moon[2], Seokwhwan Moon[2], Wonkyu Moon [1] ***

**1** Department of Mechanical Engineering, Pohang University of Science and Technology, Pohang, Gyeongbuk, Republic of Korea, **2** Department of Thoracic and Cardiovascular Surgery, Seoul St. Mary's Hospital, College of Medicine, The Catholic University of Korea, Seoul, Republic of Korea

* wkmoon@postech.ac.kr

**Data Availability Statement:** All relevant files are available from the figshare database (https://figshare.com/articles/dataset/Effects_of_precise_cardio_sounds_on_the_success_rate_of_

## Abstract

This work investigates whether inclusion of the low-frequency components of heart sounds can increase the accuracy, sensitivity and specificity of diagnosis of cardiovascular disorders. We standardized the measurement method to minimize changes in signal characteristics. We used the Continuous Wavelet Transform to analyze changing frequency characteristics over time and to allocate frequencies appropriately between the low-frequency and audible frequency bands. We used a Convolutional Neural Network (CNN) and deep-learning (DL) for image classification, and a CNN equipped with long short-term memory to enable sequential feature extraction. The accuracy of the learning model was validated using the PhysioNet 2016 CinC dataset, then we used our collected dataset to show that incorporating low-frequency components in the dataset increased the DL model's accuracy by 2% and sensitivity by 4%. Furthermore, the LSTM layer was 0.8% more accurate than the dense layer.

## Introduction

Analysis of acoustic emissions provides a convenient approach to diagnosing a system of interest by detecting sounds that it generates [1]. Sounds are usually generated by vibratory motions of a structure, so they may include some information on its dynamic behaviors. For example, the heart continuously emits sounds, which should include some information on its state. Physicians analyze signals that are detected using a stethoscope. However, the differences between the sounds from the hearts in different medical states are subtle and confusing, so this method requires an expert who has excellent hearing and considerable experience with sounds from many human beings with various heart problems. Even good cardiology fellows could diagnose heart diseases with an accuracy of only 56.2% [2]. Advances in signal-processing and artificial intelligence technologies may help people increase the ease and accuracy at of analysis of heart sounds. Hence, many studies have been performed to precisely diagnose heart beat sounds by using deep- learning (DL) and signal-processing techniques for signal analyses in time or frequency domains or a combination of them [3]. Nonetheless, these methods are apparently not mature enough for practical use. Therefore, further studies are needed.

phonocardiography/23708010, DOIs 10.6084/m9.figshare.23708010).

**Funding:** This study was supported by the 2017 Seoul St. Mary's Hospital Leading Specialization Project (KC17TESI0684, Development of the optimized sensor and the deep learning – based automated cardiac sound analysis system for digital stethoscope) funded By Seoul St. Mary's Hospital and the Korea-led K-Sensor Technology Development Program for Market Leadership (RS-2022-00154770, Next-generation piezoelectric based motion sensor platform) funded By the Ministry of Trade, Industry \& Energy(MOTIE, Korea). The funders had no role in study design, data collection and analysis, decision to publish, or preparation of the manuscript. There was no additional external funding received for this study.

**Competing interests:** The authors have declared that no competing interests exist.

Methods to visualize heart sounds could considerably help medical experts in diagnosing heart diseases. For example, phonocardiography, a technique to visualize heart sounds, has increased diagnosis accuracy from 59% to 68% [4]. However, may studies are currently focused on techniques to diagnose the state of the heart by detecting the sounds from it with a microphone and recording them digitally for computer analysis. Considerable advances have been achieved [3]. The decision accuracy on whether a subject has a heart problem has approached 98.3% [5]. An event called "Classification of Heart Sound Recordings: The PhysioNet/Computing in Cardiology Challenge 2016" Stimulated many researchers to start studies in this field [6]. At that time, 86% was the highest mean accuracy for the binary classification of the status of a subject's heart, which was achieved by using a conventional microphone for the audible-frequency band and AdaBoost and convolutional neural network (CNN) ensemble structure algorithms [7]. By 2019, the mean accuracy had risen to 94%, which was realized by adopting segmentation and a modified AlexNet algorithm [8]. Heart sounds provided in PhysioNet can be successfully classified into two categories (normal and abnormal) with a 98.3% success rate by using long short-term memory (LSTM) and segmentation based on logistic-regression hidden semi-Markov model (HSMM)-based techniques [5]. Other research groups have reported accuracies ranging from 86.6% to 93.2% without the segmentation process [9–12]. They achieved success rates of 90% using a CNN [9], 92.5% using long short-term memory (LSTM) [10], 98.3% using CNN-LSTM [11], and 86.6% using 1D CNN-LSTM [12]. Although the best accuracy is 98.3%, it is not high enough for practical applications. However, the potential usefulness has been sufficiently recognized, and further studies are warranted.

Diagnostic techniques have been advanced considerably by applying DL methods to analyses of the heart sounds in the audio-frequency band. However, in addition to audible sounds, the human heart generates infrasound, [13] which is greater than the audible content [14]. Any type of frequency content may provide important information on the specific behaviors of the heart. Hence, infrasound in should be considered when analyzing heart beat sounds. In the heart sound data in the PhysioNet database, some sets include infrasound components and some do not. However, conventional microphones or acoustic sensors can detect only audio sounds and their sensitivity toward infrasound may change with frequency; therefore, the data that include infrasound cannot be guaranteed to be accurate [15]. Consequently, for use of infrasound in diagnosis of heart diseases, the information that it conveys must be determined. The procedures should be carefully reviewed, and if necessary, appropriate methods should be introduced. After acquisition of precise heart sound data, DL methods should be applied to both data with and without infrasound content, and the results of the two analyses should be compared.

In this work, we investigate the effects of infrasound inclusion in analysis of heart sound data. We started by designing a measurement method for real heart sounds, because precise analyses and diagnosis requires acquisition of accurate signals from the subjects. Conventional audio-frequency heart sounds can be extracted from sounds that include infrasound components by using a high-pass filter, so sets of data that include and do not include infrasound signals were constructed from the acquired signals. These two groups of data were analyzed separately using the same DL techniques to diagnose the status of the heart.

## Materials and methods

### Ethics statement

We collected heart sound data from participants by using and measurement setup that we established. This work was approved by the Institutional Review Board for Seoul St. Mary's Hospital (approval no. KC17TESI0684). Participants were recruited from January 21, 2019 to

August 7, 2019. All were adult. Each was assigned a unique identification number. Written informed consent was obtained from each participant, and duplicates of these consent documents are presently maintained in a sealed state within the research laboratory at Seoul St. Mary's Hospital.

## Heart sound measurement methods

We aimed to measure "intact heart sounds", which are defined as signals that include all frequencies of the sound generated by the heart. The frequency of heart sounds is expected to be extended down to 1 Hz [14], but no commercially-available microphones can measure such low-frequency heart sounds. Therefore, we established a heart-sound measurement environment that was composed of a microphone, an amplifier, and data acquisition (DAQ) equipment (Fig 1). We used B&K's Type 4193-L-004 Reference microphone as a low-frequency acoustic microphone with a reference sensing level, B&K's Nexus conditioning amplifier Type 2690-A-0S1 as an amplifier, and NI USB-6216 BNC M-series as the DAQ system.

To assist the microphone, we designed and fabricated an acoustic adapter. We opted for an acoustic-chamber-type adapter that is about the size of a conventional stethoscope. To design the adapter, we referred to the IEC-61094–5 international standard, which describes a standard method for designing an acoustic chamber-type adapter. It had radius 25 mm and length 60 mm (S1 Fig). The chamber satisfies the IEC-61094–5 standard [16]. We designed an adapter that had a cut-off frequency of 1.8 kHz, and therefore could accurately measure low-frequency heart sounds. This acoustic adapter features a sub-adapter, which can be interchanged depending on the size of the microphone mounted on it. This design allows compatibility with the ½-inch B&K 4193 microphone and also with miniature MEMS(Micro-Electromechanical Systems) microphones and large microphones up to 1 inch in size. Several factors can influence the heart sound signal, in addition to the equipment. Other factors include including the location at which the heart sound is measured, the way that the measurement device is handled, the posture of the subject, and the time of the measurement.

To ensure sufficient network learning even with a small dataset, we established a consistent measurement method to minimize as much as possible the signal change caused by the measurement method and to include only the effect of the characteristics of the heart in the dataset. We determined the location where the microphone could be most effectively adhered to the chest and where the signal was strongest. We selected the fourth costal cartilage edge as the measurement location because it offered the best signal strength. Further, hospital patients were the main target of the measurement, so we selected the basic measurement posture in which the patient lay supine on the bed. This posture does not burden the patients during the measurement. The acoustic signal originates from the heart valves, which are situated at a distance from each other, so it may have different characteristics owing to factors such as the handling of the measurement device [17]. To avoid complications due to this effect, we ensured that the measurement device was handled consistently across all measurements and that the measurement duration was kept consistent. By standardizing the measurement method, we aimed to obtain a reliable dataset that could be used for accurate heart sound diagnosis.

**Pre-processing of data.** To appropriately process the heart sound signal (S2 Fig) and extract features from both low- and high-frequency bands, its characteristics must be understood. The frequency-dependent components can be identified using a fast Fourier transform (FFT) [18, 19], and the heart sound measured by the experimental equipment has a high-amplitude low-frequency component. FFT results (Fig 2) of the signals and noise measured using our experimental setup demonstrate that heart sounds caused by cardiac activity can be

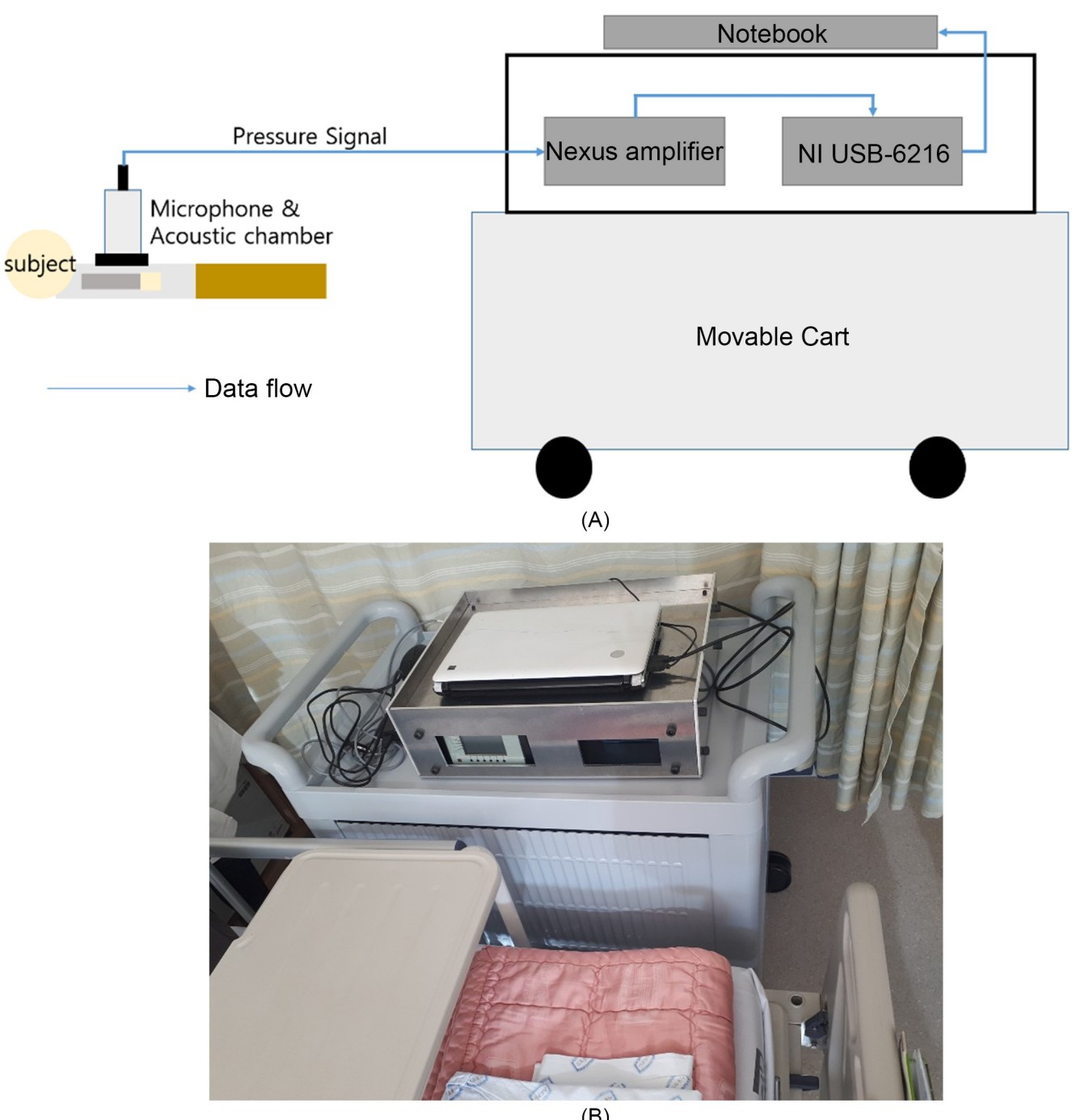

**Fig 1. Experimental device.** (A): Schematic diagram of the experimental device (B): Diagram of the experimental device installation.

detected at frequencies $\geq 1$ Hz. The Signal-to-Noise Ratio (SNR) of these heart sound signals was 33.05 dB.

To separate features of the low-frequency band separate from those of the high-frequency band, we used a time–frequency representation method, which transforms a set of time signals to their time–frequency image. The short-time Fourier transform (STFT) represents a

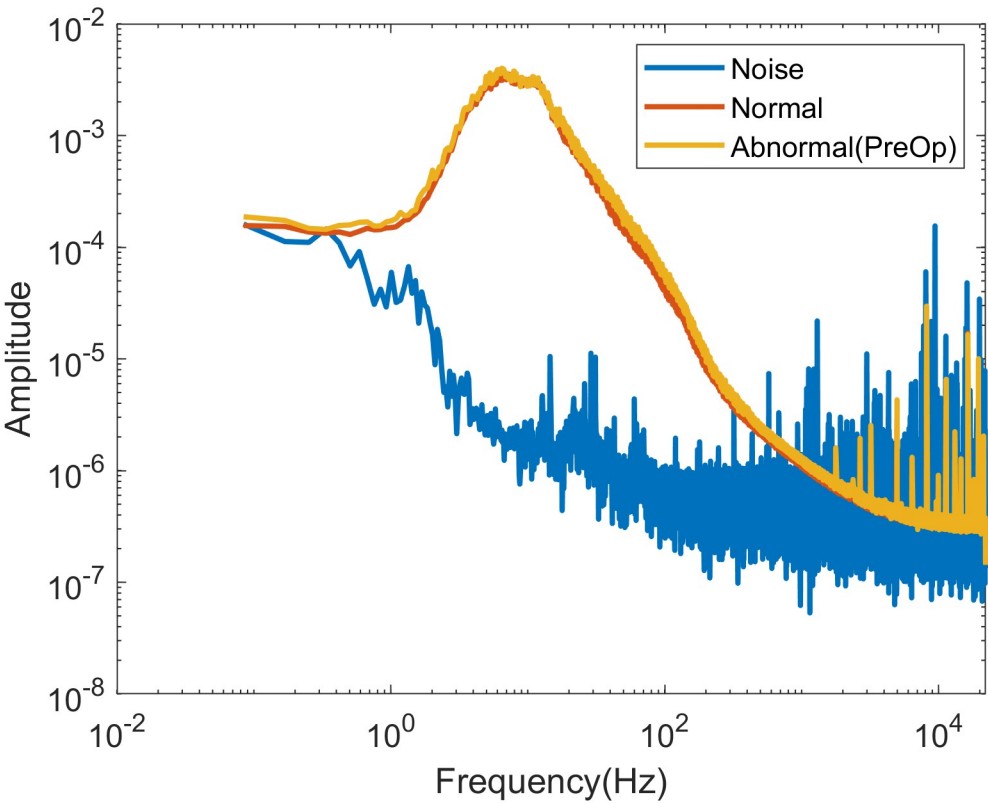

**Fig 2. FFT image of heart sound.** Average result of experiment data's FFT.

fundamental technique in signal processing. The STFT formula (Eq 1) demonstrates that the window function facilitates execution of a partial FFT concerning a short time interval. However, the formula is not well suited for extracting features from low-frequency components, primarily because it has an constrained frequency resolution [20].

$$\text{STFT}\{x(t)\}(m, \omega) = X(m, \omega) = \int_{-\infty}^{\infty} x(t)w(t - m)e^{-j\omega(t-m)} dt \tag{1}$$

Therefore, we used the continuous wavelet transform (CWT), which is defined as the integral (Eq 2) over all time of the product of the signal $f(t)$ and a scaled and translated version of the mother wavelet $\psi$. Here, $\psi_{b,a}(t)$ represents the mother wavelet (Eq 3) modified by the scale parameter $a$ and translation parameter $b$, where $a$ influences the dilation of the wavelet, allowing it to capture features at various frequencies, and $b$ adjusts the wavelet's position, facilitating the analysis of the signal's temporal characteristics [20]. This characteristic enables analysis of signal characteristics while maintaining an appropriate time–frequency resolution for each of the low- and high-frequency bands.

$$W_\psi f(b, a) := \int_{-\infty}^{\infty} f(t)\psi_{b,a}(t) dt, \tag{2}$$

$$\psi_{b,a}(t) = \frac{1}{\sqrt{a}} \psi\left(\frac{t - b}{a}\right), \quad a > 0. \tag{3}$$

STFT has a fixed time-frequency resolution, and therefore its low frequency resolution causes interference in the low-frequency band when the window is adjusted for the audible-frequency band. This distortion of the transformation results can lead to significant errors in the assessment of the effects of heart disease that is identified by low-frequency sound signals, which is a key focus of this study. In contrast, CWT allows for high-frequency resolution at low frequencies by multi-resolution, preventing distortion in the transformation results of the low-frequency range [21].

In the Wavelet Toolbox of MATLAB, a proprietary multi-paradigm programming language and numerical computing environment developed and distributed by MathWorks, the CWT function is utilized to perform the wavelet transform. The CWT function was implemented in a frequency range of 2–300 Hz, using a Morlet wavelet as the basis wavelet. This approach allowed us to convert the heart-sound signal to a reliable time–frequency representation, which could be used for subsequent analysis and feature extraction.

STFT and CWT for the heart-sound signal were obtained from the experimental equipment. Compared to the STFT image (Fig 3A), the CWT image (Fig 3B) provides a clearer representation of the frequency characteristics of the signal and enables feature extraction in both low- and high-frequency bands.

## Deep-learning network structure

This study aimed to analyze the effects of intact heart sounds on learning performance among DL models. As the basic network model, we adopted a convolutional neural net (CNN) structure, which is typically used in image classification [22]. The foundational design (Fig 4) of the CNN model was developed referencing the VGG16 model [23]. Accordingly, layer parameters were set, including the use of the same padding, and doubling of the feature size with increasing depth. Additional parameters, such as the base feature parameter of the first layer and the depth of model, were configured considering on the capabilities of the computing environment utilized. The model had a base feature of 16, a depth of four layers, and $2.03 \times 10^6$ parameters.

To incorporate the sequential nature of heart sound data, we implemented a CNN-long short-term memory (LSTM) structure (Fig 5), which is used to extract sequential information from an image, with the LSTM layer replacing the dense layer as the classification layer [24]. In healthy individuals, heartbeats are typically regular, but in ill patients, irregularities and noise can increase. The classification layer that uses LSTM can analyze regularities from such sequential data and discern features that are associated with irregularities [25]. Experimentation with CNN-LSTM enables preliminary assessment of whether sequential features occur even in low-frequency components. The model had of $5.55 \times 10^5$ parameters.

To incorporate the sequential nature of heart sound data, we implemented a CNN-LSTM structure, which is used to extract sequential information from an image, with the LSTM layer replacing the dense layer as the classification layer [24]. In healthy individuals, heartbeats are typically regular, but in cases of illness, irregularities and noise can increase. The classification layer using LSTM can analyze regularities from such sequential data and discern features associated with irregularities [25]. By experimenting with CNN-LSTM, it is possible to preliminarily assess whether there are sequential features even in low-frequency components. The CNN-LSTM model structure used for the experiment is shown in Fig 5. The model had a total of $5.55 \times 10^5$ parameters.

The hyperparameters for training were as follows: The learning rate was set to $10^{-4}$, and the batch size was set to 12 for training over 100 epochs. The Adam optimizer was used, and the loss was set to binary cross entropy. To evaluate the model, the cross-validation method [26]

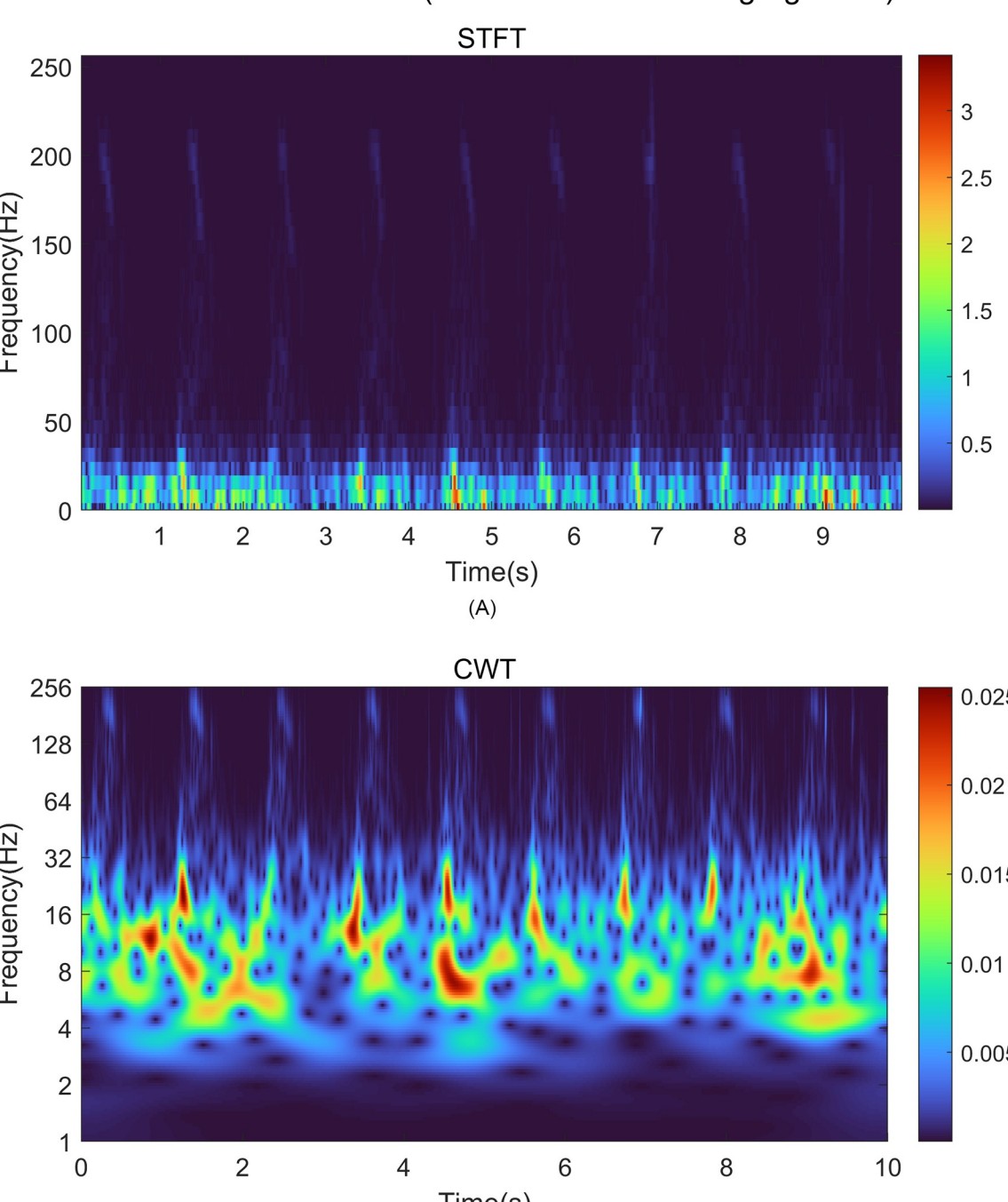

**Fig 3. Comparison of STFT and CWT for heart sound signal.** (A) STFT of the heart sound signal showing a small-size low-frequency component. (B) CWT of the heart sound signal showing a clearer representation of frequency characteristics.

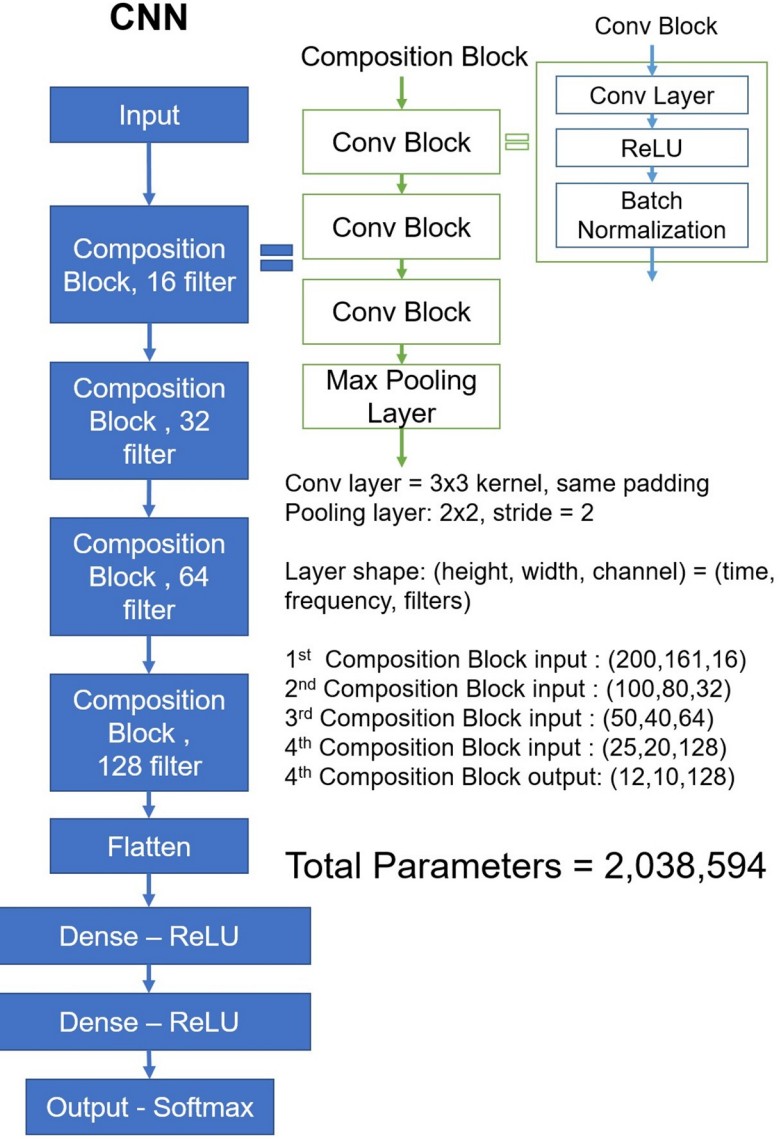

**Fig 4. CNN model structure.**

was used, and five-fold cross-validation was performed. The network structure was implemented and trained using the Keras interface in TensorFlow. The computer environment had an AMD Ryzen 1900X CPU, RAM of DDR3 1333 MHz 128 GB, and an NVDIA GEFORCE GTX 1080 Ti GPU.

## Training dataset

**Evaluation of model performance.** To accurately compare the effect of input data on diagnostic accuracy, the DL model developed here must itself have satisfactory diagnostic accuracy. Further, to validate this accuracy, a verified open dataset is required. The 2016 PhysioNet/CinC dataset is a valuable resource for developing and evaluating heart-sound diagnosis models [15]. It contains a large number of heart sound recordings, which were obtained from

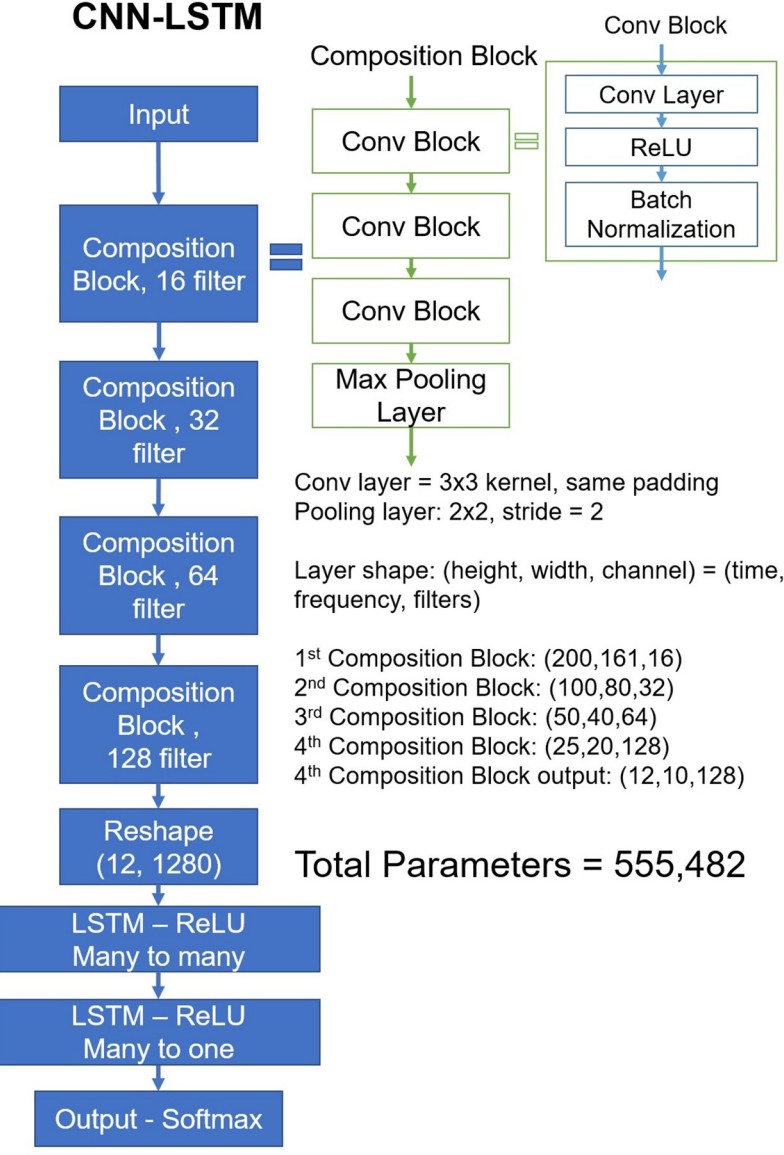

**Fig 5. CNN-LSTM model structure.**

different sources, including public databases and hospitals; hence, it provides a diverse set of data that can be used to train and test models to distinguish between binary classes (here, Normal vs. Abnormal).

However, the variability in the characteristics of the data can pose a challenge in developing accurate models. The recordings were obtained using different types of measurement equipment, different methods, and in different environments, and these differences can affect the quality and characteristics of the heart-sound signal. Additionally, the dataset contains recordings from patients of different ages, genders, and health conditions, and these differences could also influence the characteristics of the data. Despite these challenges, several studies used the PhysioNet 2016 dataset to evaluate the diagnostic accuracy of heart-sound diagnosis models. These studies have demonstrated that DL models may accurately diagnose heart

conditions by using heart-sound recordings. However, further research is required to develop models that can generalize well across different datasets and populations.

**Testing the effects of infrasound.** To quantify the effect of low-frequency heart sounds on the accuracy of diagnosis, two datasets are needed: one that includes low-frequency components and one that omits them. Also, the characteristics of heart sounds depend on the measurement location, so measurements must be taken at the same location. is necessary. As the number of microphones increases, the size constraint of the adapter intensifies, so use multiple microphones is restricted by the curvature of the chest. To solve this problem, a dataset including low-frequency components was constructed using a single microphone, and a dataset without low-frequency components was generated by using software to eliminate them. To create the comparison dataset without low-frequency components, a high-pass filter with a cut-off frequency of 20 Hz was applied to the original dataset. This process removed all frequency components below 20 Hz, and left only audible-frequency components. The high-pass filter was designed using the 'filter-design' tool in MATLAB, then applied with phase correction using the 'filter' function in MATLAB to ensure that the filtering process did not add phase delay to the filtered signal. The effect of low-frequency components on the diagnosis of cardiovascular diseases was then evaluated by comparing learning results obtained using the original dataset that included low-frequency components to the results obtained using the dataset that lacked low-frequency components. This comparison was achieved by training the same network model on both datasets and comparing their Accuracies, Specificities, and Sensitivities on a test set.

## Results and discussion

### Collected dataset

We used the heart sound measurement method established for experimentation to we collect heart-sound data at Seoul St. Mary's Hospital of the Catholic University, and created a dataset (Table 1). The data were collected from three distinct groups: a pre-operative group composed of individuals waiting for surgery for heart valve abnormalities, a post-operative group composed of subjects who had undergone surgery, and a normal group consisting of participants who had no cardiovascular abnormalities. The recordings of the individuals who had heart diseases were assigned to three classes: 'primary', 'secondary 1', and 'secondary 2', which had been diagnosed by a cardiovascular specialist using echocardiography imaging.

The primary disease among the subjects was cardiovascular abnormality, a condition typically diagnosed by experts using ultrasound imaging.

This experiment provides valuable insights into the heart sounds of individuals with various heart diseases and offers a comprehensive dataset for future research in the field. The use of an identification system also ensures that the data can be effectively organized and analyzed; this

**Table 1. Dataset statistics.**

|  | Normal | Pre-op | Post-op | Total |
|---|---|---|---|---|
| **Subject** | 85 | 62 | 51 | 198 |
| **Sex** | M: 52 | M: 31 | M: 22 | M: 105 |
|  | F: 33 | F: 31 | F: 29 | F: 93 |
| **Age** | 65.68 ± 10.87 | 65.59 ± 10.89 | 65.58 ± 10.88 | 65.65 ± 10.82 |
| **Data number** | 276 | 195 | 177 | 648 |

Statistics of collected dataset

ability is crucial for increasing the understanding of these conditions and developing effective treatment strategies.

## Model validation result

In this experiment, the diagnostic accuracies of the CNN and CNN-LSTM models were evaluated using the PhysioNet dataset. The models were tested using different configurations, including applying the LSTM layer instead of the classification layer with the dense layer. The results (Table 2) showed that use of the LSTM layer increased accuracy by 0.5% compared to the baseline configuration. Both models achieved an accuracy ≥ 92%; i.e., the model design was effective and well-executed. The use of the PhysioNet dataset in this experiment facilitated evaluation of model accuracy in detecting and classifying heart-related abnormalities. The high accuracy achieved by both models suggests that they are well suited for use in real-world scenarios in which accurate diagnosis is crucial. These results demonstrate the potential of using DL techniques such as CNN and CNN-LSTM in medical applications, particularly in cardiology. Overall, this experiment demonstrated the effectiveness of incorporating LSTM layers in CNN models for classification of heart-related abnormalities. The high accuracy achieved by both models suggests that they could be useful tools in assisting medical professionals in the accurate diagnosis of cardiac conditions.

The training accuracy the models used in our study were 92.65% for the CNN-dense model and 93.16% for the CNN-LSTM model. Both sufficiently validate the high quality of the model. These accuracies are higher than those of the CNN model reported in 2019 [9], but lower than the CNN-LSTM [11] model, which achieved a training accuracy of 98.34%; the increased was attributed to an improvement of 2% by using dropout. Also, this model [11] uses a feature size in the LSTM layer that was 5.3 times greater than in our model. Additionally [11] included one additional dense layer before the final softmax layer. Although we assumed that two LSTM layers would be sufficient for extracting sequential features and functioning effectively as a classification layer, the lower accuracy compared to LSTM-dense suggests that LSTM might be less effective than dense layers for classification.

## Train result

The network model was trained using the collected dataset, and its diagnostic performance was evaluated using accuracy, sensitivity, and specificity as indicators. The classification classes were defined as binary classes, where the 'Positive' group was designated from the dataset

**Table 2. Model evaluation results.**

| Model | Accuracy(%) | Sensitivity(%) | Specificity(%) |
|---|---|---|---|
| Adaboost+CNN(2016) [7] | 86.02 | 94.24 | 77.81 |
| Modified AlexNet(2017) [8] | 94.16 | 95.12 | 93.20 |
| BLSTM(2018) [5] | 97.63 | 98.86 | 98.36 |
| CNN(2019) [9] | 90 | 90 | 90 |
| LSTM(2019) [10] | 92.35±1.09 | 94.22±2.41 | 90.48±1.74 |
| CNN-LSTM(2020) [11] | 98.34 | 98.66 | 98.01 |
| 1D CNN-LSTM(2020) [12] | 86.61 | 59.06 | 93.31 |
| **CNN-dense** | **92.65 ± 0.48** | **86.63 ± 1.56** | **94.58 ± 0.68** |
| **CNN-LSTM** | **93.16 ± 0.66** | **87.88 ± 2.14** | **94.87 ± 1.04** |

Comparison of the training results of PhysioNet datasets

**Table 3. Training results.**

| Model | Dataset | Accuracy(%) | Sensitivity(%) | Specificity(%) |
|---|---|---|---|---|
| **CNN-dense** | *Audible* | 85.64 ± 2.52 | 74.06 ± 8.26 | 92.25 ± 1.68 |
| | *Infrasound* | 88.80 ± 2.14 | 78.62 ± 5.18 | 94.63 ± 3.75 |
| | *Difference* | **3.16** | 4.56 | **2.38** |
| **CNN-LSTM** | *Audible* | 87.42 ± 2.99 | 72.75 ± 8.68 | **95.81 ± 0.79** |
| | *Infrasound* | **89.64 ± 3.68** | **79.76 ± 9.02** | 95.26 ± 2.47 |
| | *Difference* | 2.22 | **7.01** | −0.55 |

Comparison of training results of heart sound datasets converted with CWT

labeled as 'Abnormal (Pre-op)', and the 'Negative' group was established from the dataset comprising 'Normal subjects'.

The results (Table 3) show that the CNN-DNSE model achieved 3.16% higher accuracy when it used the dataset that included infrasound than when it used the dataset that included only audible data. Use of the set that included ultrasound increased the sensitivity by 4.56% and the specificity by 2.38%. Similarly, the CNN-LSTM model achieved 2.22% higher accuracy when it used the dataset that included infrasound than when it used the dataset that included only audible data, and use the set that included ultrasound increased the sensitivity by 7.01%; however, it decreased the specificity by 0.55. The CNN-LSTM model showed 1.8% higher accuracy than the CNN-dense model. This result suggests that the use of infrasound data could improve the diagnostic accuracy of both models, particularly in terms of sensitivity. In the cross-validation results, CNN and CNN-LSTM models show a higher standard deviation in accuracy when trained on the data collected for this study (2.76%, on the dataset that included infrasound, and 2.91% on the dataset that excluded it), than when trained with the PhysioNet dataset (0.48%, on the dataset that included infrasound, and 0.65% on the dataset that excluded it). Increased standard deviation in cross-validation suggests an increased tendency towards overfitting; the likelihood of this phenomenon decreases as the dataset size increases, so these results suggest that the dataset size collected in this experiment may not be sufficiently large. Additionally, our models gave consistently higher standard deviation of Sensitivity than of Specificity. This result could be attributed to slight imbalance in the dataset, which included 1.41 times more normal data than abnormal data. CNN-LSTM showed an overall increase in the average and decrease in standard deviation of Specificity compared to CNN-dense. However, for Sensitivity, use of the dataset that excluded infrasound achieved lower average value and a similar standard deviation, whereas use of the dataset that included infrasound, the average increased but the standard deviation also increased. This analysis suggests that although LSTM's sequential feature facilitated identification of regular normal patterns, it appears to have had little effect on the diagnosis of heart sounds in abnormal subjects. Overall, the results of this study demonstrate the potential of DL models in diagnosing heart-related abnormalities. The use of different types of wide-frequency-band measurements, such as infrasound, could lead to improved diagnostic accuracy, particularly in cases where use of traditional diagnostic stethoscope may not be sufficient.

## Conclusion

This study conducted experiments to investigate whether including low-frequency component data in heart sound analysis can increase diagnostic accuracy. The results showed that including low-frequency component data in the dataset increased the diagnostic Accuracy of the DL

model by an average of 2% increased its Sensitivity by an average of 4%. The study also compared the accuracies of models that LSTM to those that used dense layers as the classification layer, and found that the LSTM model was 0.8% more accurate. The study also revealed that commercially-available microphones are unsuitable for measuring low-frequency heart sounds. This result indicates a need for specialized equipment that can effectively measure and analyze low-frequency heart sounds. Finally, the findings of this study provide valuable insights into the potential of using low-frequency component data in heart-sound analysis and the importance of specialized equipment for accurate measurements. The results have practical implications for healthcare professionals who rely on accurate diagnostic tools to detect and treat heart-related abnormalities. Further research could involve benchmarking a variety of models. Different preprocessing techniques or models, such as those other than the currently-used CWT, CNN-dense, or CNN-LSTM, may achieve higher accuracy than those that do use them. Understanding can be extended by testing various advanced models after expanding the dataset. Use of the experimental equipment, which has flat frequency characteristics in infrasound, confirmed the diagnostic potential of low-frequency heart sounds. However, the experimental equipment is relatively large, and this characteristic may impede their use. Hence, clinical adoption of infrasound will require development of new ultra-compact infrasound sensors.

## Supporting information

**S1 Fig. ARTA acoustic chamber model & result.** The ARTA acoustic chamber model design method is used to design acoustic adapters that can accurately measure sound pressure in an experimental setting. One important parameter in this design method is the cut-off frequency, which is defined as the frequency at which the sound pressure has a difference of 1 dB per cm displacement in the longitudinal direction. If a detected frequency exceeds the cut-off frequency, it causes a measurement error. Therefore, the designed acoustic adapter must have a cut-off frequency that exceeds the maximum frequency of the sound being measured. In the case of heart sounds, the maximum frequency is known to be 1 kHz. To design the acoustic adapter, finite element modeling was used for calculation according to the design method. A COMSOL model was created, and the sound pressure difference between Point 1, where the microphone is located, and Point 2, which is 1 cm away from Point 1, was calculated. The frequency above which the sound pressure difference exceeds 1 dB becomes the cut-off frequency of the acoustic adapter. By following this design method, geometric conditions that satisfy the required cut-off frequency can be found and designed. This result ensures that the acoustic adapter can accurately measure sound pressure in the experimental setting; this ability is important for increasing the accuracy and effectiveness of diagnostic tools for cardiovascular diseases.
(TIF)

**S2 Fig. Time domain image.** Example of time domain data.
(TIF)

**S1 Dataset. Infrasound heart sound dataset repository.** https://doi.org/10.6084/m9.figshare.23708010.
(URL)

## Author Contributions

**Conceptualization:** Youngsin Kim, Mihyung Moon, Seokwhwan Moon, Wonkyu Moon.

**Data curation:** Mihyung Moon, Seokwhwan Moon.

**Formal analysis:** Youngsin Kim, Wonkyu Moon.

**Funding acquisition:** Mihyung Moon, Seokwhwan Moon, Wonkyu Moon.

**Investigation:** Youngsin Kim, Mihyung Moon, Seokwhwan Moon.

**Methodology:** Youngsin Kim, Wonkyu Moon.

**Project administration:** Seokwhwan Moon, Wonkyu Moon.

**Resources:** Youngsin Kim, Mihyung Moon, Seokwhwan Moon, Wonkyu Moon.

**Software:** Youngsin Kim, Wonkyu Moon.

**Supervision:** Mihyung Moon, Seokwhwan Moon, Wonkyu Moon.

**Validation:** Youngsin Kim, Mihyung Moon, Wonkyu Moon.

**Visualization:** Youngsin Kim.

**Writing – original draft:** Youngsin Kim, Wonkyu Moon.

**Writing – review & editing:** Mihyung Moon, Seokwhwan Moon, Wonkyu Moon.

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
