## [Decision Letter · Decision Letter 0]

21 Nov 2023

PONE-D-23-23289Effects of precise cardio sounds on the success rate of phonocardiographyPLOS ONE

Dear Dr. Moon,

Thank you for submitting your manuscript to PLOS ONE. After careful consideration, we feel that it has merit but does not fully meet PLOS ONE’s publication criteria as it currently stands. Therefore, we invite you to submit a revised version of the manuscript that addresses the points raised during the review process.

We look forward to receiving your revised manuscript.

Kind regards,

Ali Mohammad Alqudah

Academic Editor

PLOS ONE

Journal Requirements:

3, Thank you for stating in your Funding Statement:

‘This study was supported by the Korea-led K-Sensor Technology Development Program for Market Leadership (RS-2022-00154770, Next-generation piezoelectric based motion sensor platform) funded By the Ministry of Trade, Industry \\& Energy(MOTIE, Korea). The funders had no role in study design, data collection and analysis, decision to publish, or preparation of the manuscript.”

Reviewers' comments:

Reviewer's Responses to Questions

**Comments to the Author**

1. Is the manuscript technically sound, and do the data support the conclusions?

Reviewer #1: Partly

Reviewer #2: Partly

2. Has the statistical analysis been performed appropriately and rigorously? 

Reviewer #1: No

Reviewer #2: No

3. Have the authors made all data underlying the findings in their manuscript fully available?

Reviewer #1: No

Reviewer #2: Yes

4. Is the manuscript presented in an intelligible fashion and written in standard English?

Reviewer #1: Yes

Reviewer #2: Yes

5. Review Comments to the Author

Reviewer #1: "In this paper, the authors investigate the impact of low-frequency components, referred to as 'infrasound,' on the accuracy of classifying PCG (Phonocardiogram) sounds. The approach is novel and intriguing, though the authors need to establish, theoretically or clinically, whether infrasound components are directly related to the diagnosis of medical conditions. Nevertheless, the study provides valuable insights, suggesting that exploring this avenue is worthwhile.

To bolster their conclusions in this direction, the authors should design experiments and analyze the dataset more systematically, encouraging the research community to pursue this approach. For instance, the authors need to demonstrate that their 'infrasound' approach outperforms the one based on the widely known PhysioNet dataset. This could involve benchmarking the learning with various models, as it's possible that models other than CNN-Dense and CNN-LSTM may achieve better performance.

Additionally, the authors should consider conducting post-analysis on their dataset, especially in terms of evaluating the Signal-to-Noise Ratio (SNR) of the samples. This is important because they are exploring new spectral low-frequency features of heart sounds, which may or may not be as noisy as audible sounds. please check this reference:

Barnawi, Ahmed, Mehrez Boulares, and Rim Somai. 2023. "Simple and Powerful PCG Classification Method Based on Selection and Transfer Learning for Precision Medicine Application" Bioengineering 10, no. 3: 294. https://doi.org/10.3390/bioengineering10030294

Finally, the authors should revisit the preprocessing phase, specifically the conversion of sound into images, and ensure that Short-Time Fourier Transform (STFT) and Continuous Wavelet Transform (CWT) do not compromise the low-frequency components during this conversion. We have doubts about this because of two reasons: firstly, Figure 2 shows that components below 1Hz appear flat, and secondly, these conversions are intended to mimic the human perception of audible sound via aggressive filtering of irrelavant sound componants."

Reviewer #2: The paper is well-presented, but there are several considerations that need attention:

The rationale behind selecting STFT and CWT for data transformation remains undocumented. It would be beneficial to elaborate on the choice of these two transforms.

Information regarding the classification classes isn't explicitly stated. This includes the number of classes and their names, which should be clarified.

Details about the dataset used for training, specifically the quantity of samples and the data split (e.g., 70% for training and 30% for testing), are missing.

Utilizing the extensive PhysioNet dataset (with 18 classes) raises questions about how many classes were actually chosen for validation.

The process for parameter selection in each layer of the model should be explained to provide a clearer understanding.

Elucidate the advantages of using LSTM over employing a softmax classifier or feature extraction followed by a traditional machine learning classifier.

Further elaboration on the results would be beneficial to enhance clarity and provide a deeper analysis.

Considerations about dealing directly with signals rather than transforming them into images should be addressed.

Address the comparison of your results with existing studies where your achieved performance might not be the highest, providing context and potential reasons for variations.

Regarding your designed circuits, clarifications regarding novelty and their compatibility with existing microphones would be beneficial for understanding their applicability.

6. PLOS authors have the option to publish the peer review history of their article (what does this mean?). If published, this will include your full peer review and any attached files.

Reviewer #1: No

Reviewer #2: No

---

## [Author Response · Author response to Decision Letter 0]

9 Jan 2024

Responses to the Associate Editor’s and Reviewer’s Comments

05 January 2024

Dear reviewers and editorial staffs in PLOS ONE

We are profoundly grateful for the insightful critiques and constructive suggestions provided on our manuscript titled “Effects of precise cardio sounds on the success rate of phonocardiography”, control number PONE-D-23-23289. These comments have been invaluable in enhancing the scholarly rigor and clarity of our work. In recognition of the meticulous efforts of both reviewers and editors, we have endeavored to refine our manuscript according to the provided suggestions. We believe these enhancements have notably improved the scientific and clinical rigor of our manuscript. We hope that our revised manuscript will be considered and accepted for publication in PLOS ONE. We eagerly anticipate the possibility of our work contributing to the esteemed journal.

In the revised version, we have highlighted all changes for your convenience; these are underlined and colored in blue. This is to ensure that our revisions are easily identifiable and can be efficiently evaluated.

Reviewer #1:

<Review Comments>

1) Reviewer’s comment: To bolster their conclusions in this direction, the authors should design experiments and analyze the dataset more systematically, encouraging the research community to pursue this approach. For instance, the authors need to demonstrate that their 'infrasound' approach outperforms the one based on the widely known PhysioNet dataset. This could involve benchmarking the learning with various models, as it's possible that models other than CNN-Dense and CNN-LSTM may achieve better performance.

Author’s response: 

If we intend to prove that every diagnosis must be done by use of the true heart sound including its infrasound content, we have to complete the works suggested by Reviewer 1 in the above comments. However, our conclusions are different from the previous statement. 

What we claim in the manuscript may be summarized as follows: 

1. The infrasound contents should be included in the data for the diagnoses on heart diseases since a lot of infrasound contents are included in the real heart sounds in addition to the audible ones. 

2. The true heart sound (audio + infrasound) data provided with the better diagnostic results with several typical learning models.

3. The two reasons shown above encourage the researchers in the heart disease diagnosis using heart sounds and Deep Learning models.

We believe that what Reviewer 1 request us to do could be done through a lot of long-term research works that cannot and need not be done by our work in the manuscript. 

2) Reviewer’s comment: Additionally, the authors should consider conducting post-analysis on their dataset, especially in terms of evaluating the Signal-to-Noise Ratio (SNR) of the samples. This is important because they are exploring new spectral low-frequency features of heart sounds, which may or may not be as noisy as audible sounds. please check this reference:

 Barnawi, Ahmed, Mehrez Boulares, and Rim Somai. 2023. "Simple and Powerful PCG Classification Method Based on Selection and Transfer Learning for Precision Medicine Application" Bioengineering 10, no. 3: 294. 

https://doi.org/10.3390/bioengineering10030294

Author’s response: 

We deeply agree that the SNR of the true heart sound data should be checked if any noise but the heart sounds should be excluded. We have checked this already by we did not present it in the manuscript. However, as Reviewer 1 recommended, we had better show the results in the revised manuscript. Please see Line 129 and Figure 2 in the revised manuscript.

Revised Manuscript: 

 Figure 2 presents the FFT results of the signals and noise measured using the experimental setup constructed in this study. It demonstrates that heart sounds caused by cardiac activity can be detected from 1 Hz above. The Signal-to-Noise Ratio (SNR) of these heart sound signals is 33.05 dB. (LINE 129 – 132)

3) Reviewer’s comment: the authors should revisit the preprocessing phase, specifically the conversion of sound into images, and ensure that Short-Time Fourier Transform (STFT) and Continuous Wavelet Transform (CWT) do not compromise the low-frequency components during this conversion. We have doubts about this because of two reasons: firstly, Figure 2 shows that components below 1Hz appear flat, and secondly, these conversions are intended to mimic the human perception of audible sound via aggressive filtering of irrelavant sound componants.

Author’s response: 

We have ensured both the CWT or the STFT would not compromise the low-frequency components at all. The flat frequency contents below 1 Hz in Figure 2 of the original manuscript are shown to come from the internal noise of the instruments used in the experiments. This point can be clearly checked in Figure 2 of the revised manuscript. 

Moreover, we don’t agree the last claim of Reviewer 1 because the CWT or STFT is not such a thing. The Fourier and the Continuous Wavelet transforms are the mathematical transform from a function of time to another one of frequency. The relationships between the original functions and the transformed ones may be that the transformed ones would be harmonic decompositions of the original ones. Of course, their digital versions, such as FFTs, may change the relationship more complicated but the validity of the ideal relationship can be guaranteed if proper values of time intervals and the total periods of signal time series data are selected. Please see the textbooks, The Fast Fourier Transform and Its Applications [19,20] for the Fourier transform and Fundamentals of Wavelets [21] for the Continuous Wavelet transform.

[19] Brigham, E. O. (1988). The Fast Fourier Transform (FFT). In The Fast Fourier Transform and Its Applications (Chapter 9). Prentice Hall.

[20] Brigham, E. O. (1988). FFT Transform Applications. In The Fast Fourier Transform and Its Applications (Chapter 10). Prentice Hall.

[21] Goswami, J.C. and Chan, A.K. (2011). Time-Frequency Analysis. In Fundamentals of Wavelets (eds K. Chang, J.C. Goswami and A.K. Chan).

https://doi.org/10.1002/9780470926994.ch4

Reviewer #2 :

<Review Comments>

1) Reviewer’s comment: The rationale behind selecting STFT and CWT for data transformation remains undocumented. It would be beneficial to elaborate on the choice of these two transforms.

Author’s response: 

In this study, we elaborate on the reasons for using Short-Time Fourier Transform (STFT) and Continuous Wavelet Transform (CWT) for data transformation. The primary reason for choosing CWT is its capability for multi-resolution analysis, allowing for maintaining consistent resolution in both infrasound and audible frequency ranges. CWT employs logarithmic frequency resolution in octaves, enabling the transformation of low and regular frequencies into images within an equivalent domain. This is particularly beneficial for maintaining high resolution in the low-frequency domain and effectively separating signal from high low-frequency noise. On the other hand, STFT uses a fixed resolution and is typically set for time-frequency resolutions suited for the audible frequency band. This presents challenges in fitting low-frequency frequency resolution adequately. 

 based on these technical considerations, CWT was selected as the primary method of data transformation in this study, and this rationale has been added to the research methodology section.

Revised Manuscript:

STFT, due to its fixed time-frequency resolution, causes interference in the low-frequency band due to low frequency resolution when the window is adjusted for the audible frequency band. This distortion of the transformation results can lead to significant errors in the assessment of the impact of low-frequency heart diseases, which is a key focus of this study. In contrast, CWT allows for high-frequency resolution in low frequencies through multi-resolution, preventing distortion in the transformation results of the low-frequency range. [22] (LINE 148 - 154)

[22] Graps, A. (1995). An introduction to wavelets. IEEE computational science and engineering, 2(2), 50-61.

2) Reviewer’s comment: Information regarding the classification classes isn't explicitly stated. This includes the number of classes and their names, which should be clarified.

Author’s response: 

Recognizing the importance of providing clear information about classification classes, we have enhanced the content in the Results section. The dataset collected in this study was gathered from various groups of subjects, including patients with different diseases, patients with varying intensities of a particular disease, and post-operative patients.

However, for the classification task in this study, we adopted a simplified approach to reduce the complexity of the dataset and enhance the clarity of the research. Specifically, the classification system was set dichotomously, consisting of two main classes: 'Pre-operative subjects' (Positive class) and 'Normal subjects' (Negative class). Data from post-operative patients were excluded from this classification task. We believe this approach helps focus on the core objectives of our study and makes the interpretation of results clearer.

Revised Manuscript:

 The classification classes were defined as binary classes, where the 'Positive' group was designated from the dataset labeled as 'Abnormal (Pre-op)', and the 'Negative' group was established from the dataset comprising 'Normal subjects'. (Line 293 - 296)

3) Reviewer’s comment: Details about the dataset used for training, specifically the quantity of samples and the data split (e.g., 70% for training and 30% for testing), are missing.

Author’s response: 

In this study, instead of the standard 7:3 (70% training, 30% testing) data split, we utilized 5-fold cross-validation to divide the data into 80% for training and 20% for validation. The primary reason for this decision was our assessment that the size of the dataset we collected was not yet sufficiently large.

Particularly in cases where the dataset is small, as with ours, the conventional 7:3 split can lead to significant variations in model performance evaluation, depending on the selection of the test set. As observed in the results of our paper, the performance variability when using our dataset was much more significant compared to the larger PhysioNet dataset. This variability can be several times greater depending on the size of the dataset.

The 5-fold cross-validation method was chosen in consideration of these dataset limitations. This approach allows for a more stable evaluation of the model's learning performance and helps reduce the risk of overfitting, especially in small datasets. [27] This decision was made after consulting relevant studies and plays a crucial role in enhancing the reliability of our research results. 

Revised Manuscript:

To evaluate the model, the cross-validation method [27] was used, and 5-fold cross-validation was performed in the experiment. (Line 188 – 190)

[27] Goodfellow, I., Bengio, Y., & Courville, A. (2016). Machine Learning Basics. In Deep Learning (pp. 121-123). MIT Press.

4) Reviewer’s comment: Utilizing the extensive PhysioNet dataset (with 18 classes) raises questions about how many classes were actually chosen for validation

Author’s response: 

In our study, we utilized the 2016 PhysioNet/CinC Challenge dataset, which primarily offers two dichotomous classes: 'Normal' and 'Abnormal'. The focus of our study is on distinguishing between these two classes, and as such, all analyses and modeling were conducted based on these two categories.

Consequently, we carried out data labeling and one-hot encoding processes centered on these two classes. This approach aligns with the objective of our study, which is to differentiate between normal and abnormal heart sounds, thereby enhancing the clarity of results and ease of interpretation. Therefore, among the various classes provided by the PhysioNet dataset, only these two classes were utilized in our study.

Revised Manuscript:

The 2016 PhysioNet/CinC dataset is a valuable resource for developing and evaluating heart sound diagnosis models.[16] It contains a large number of heart sound recordings, which were obtained from different sources, including public databases and hospitals; hence, it provides a diverse set of data to train and test models with binary classes (Normal, Abnormal). (Line 200 - 204)

5) Reviewer’s comment: The process for parameter selection in each layer of the model should be explained to provide a clearer understanding

Author’s response: 

The model used in our study was inspired by the structure of the VGG model, adopting key parameters of the VGG model such as 'same padding', 'feature size doubling', and '1/2 max-pooling'. 

However, the major differences from the VGG model lie in the depth of each convolutional layer, the size of the base feature, and the number of features in the dense layers. The selection of these parameters was primarily influenced by the limitations of RAM and VRAM in our team's computing environment. Generally, an increased number of features can improve performance, but it also requires more memory due to the higher number of parameters, leading to hardware constraints.

Therefore, in our study, we chose the optimal values that our computing environment could accommodate for designing our model. We then validated this custom-designed model using the PhysioNet dataset, comparing its performance with existing studies to ascertain if our model demonstrates similar effectiveness. This validation process played a crucial role in verifying the model's feasibility and enhancing the reliability of our research findings.

Revised Manuscript:

The foundational design of the CNN model was developed referencing the VGG16 model [24]. Accordingly, layer parameters were set, including the use of same padding and doubling the feature size with increasing depth. Additional parameters, such as the base feature parameter of the first layer and the depth of model, were configured based on the performance capabilities of the computing environment utilized. (LINE 169 – 174)

The computer environment employed comprised a CPU AMD Ryzen 1900X, RAM of DDR3 1333MHz 128 GB, and a GPU NVDIA GEFORCE GTX 1080 Ti (LINE 191 - 193)

[24] Simonyan, K., & Zisserman, A. (2014). Very deep convolutional networks for large-scale image recognition. arXiv preprint arXiv:1409.1556.

6) Reviewer’s comment: Elucidate the advantages of using LSTM over employing a softmax classifier or feature extraction followed by a traditional machine learning classifier.

Author’s response: 

We'll clearly explain the primary reasons for using Long Short-Term Memory (LSTM) in our study. Firstly, it's important to note that a softmax classifier was also employed in our CNN-LSTM model. However, there is a significant difference between the approach of traditional machine learning classifiers or feature extraction in Dense layers and the approach of LSTM.

Dense layers primarily extract local features, which reflect fixed patterns in images or time-series data. In contrast, LSTM layers focus on analyzing sequential characteristics such as the continuity or repetitiveness of these local features.[26] Heart sound data inherently possesses characteristics reflecting the regular heartbeat patterns, and these regularities are crucial for identifying abnormal heart sound patterns. LSTM is adept at effectively analyzing and modeling the temporal characteristics and regularities of such sequence data. This is particularly effective for data like heart sounds where temporal continuity is significant, and we believe it is one of the reasons why our CNN-LSTM model achieved high accuracy with fewer parameters. This method of analysis is evidenced in the results of our study, demonstrating that the use of LSTM in processing heart sound data offers superior performance compared to traditional machine learning methods.

Revised Manuscript:

In healthy individuals, heartbeats are typically regular, but in cases of illness, irregularities and noise can increase. The classification layer using LSTM can analyze regularities from such sequential

---

## [Decision Letter · Decision Letter 1]

1 Feb 2024

PONE-D-23-23289R1Effects of precise cardio sounds on the success rate of phonocardiographyPLOS ONE

Dear Dr. Moon,

Thank you for submitting your manuscript to PLOS ONE. After careful consideration, we feel that it has merit but does not fully meet PLOS ONE’s publication criteria as it currently stands. Therefore, we invite you to submit a revised version of the manuscript that addresses the points raised during the review process.

We look forward to receiving your revised manuscript.

Kind regards,

Ali Mohammad Alqudah

Academic Editor

PLOS ONE

Reviewers' comments:

Reviewer's Responses to Questions

**Comments to the Author**

1. If the authors have adequately addressed your comments raised in a previous round of review and you feel that this manuscript is now acceptable for publication, you may indicate that here to bypass the “Comments to the Author” section, enter your conflict of interest statement in the “Confidential to Editor” section, and submit your "Accept" recommendation.

Reviewer #1: (No Response)

2. Is the manuscript technically sound, and do the data support the conclusions?

Reviewer #1: Partly

3. Has the statistical analysis been performed appropriately and rigorously? 

Reviewer #1: N/A

4. Have the authors made all data underlying the findings in their manuscript fully available?

Reviewer #1: No

5. Is the manuscript presented in an intelligible fashion and written in standard English?

Reviewer #1: Yes

6. Review Comments to the Author

Reviewer #1: the authors didn't respond to the first comment that they need to establish some clinical relevance to infrasound and cardiac diagnosis. also the response to the other comments were unsatisfactory.

7. PLOS authors have the option to publish the peer review history of their article (what does this mean?). If published, this will include your full peer review and any attached files.

Reviewer #1: No

---

## [Author Response · Author response to Decision Letter 1]

23 Feb 2024

Responses to Reviewer’s Comments

February 23, 2024

Dear reviewer in PLOS ONE

Reviewer #1:

<Review Comments>

1) Reviewer’s comment: the authors didn't respond to the first comment that they need to establish some clinical relevance to infrasound and cardiac diagnosis. 

First comments in the previous review: In this paper, the authors investigate the impact of low-frequency components, referred to as 'infrasound,' on the accuracy of classifying PCG (Phonocardiogram) sounds. The approach is novel and intriguing, though the authors need to establish, theoretically or clinically, whether infrasound components are directly related to the diagnosis of medical conditions. Nevertheless, the study provides valuable insights, suggesting that exploring this avenue is worthwhile.

Author’s response: 

We regret we did not provide our specific answer to the first comment of Reviewer 1. However, due to the last sentence of his/her comments given above, we thought we don’t have to respond to the comments specifically. Moreover, since our answers to the second comments of Reviewer 1 includes our answers to the first comments implicitly, we have missed the step required by the reviewer. 

As is requested in the first sentence of the current comments from Reviewer 1, our specific responses to the first comments would be as follows: 

As in our previous responses, we believe that we have already ‘theoretically’ and experimentally established whether infrasound components are directly related to the heart conditions in the previous first revision. It is shown that the detected sounds including the infra-frequency components are from the heart, (Our experimental procedures for acquisition of heart sounds would prove it.) hence, they could be highly related to the physical conditions of heart. This shows the relationship between heart sounds including infra-frequency components and the heart condition. Since Reviewer 1 requests to establish, ‘theoretically or clinically’, we have almost done his/her request already. 

The remaining may be to establish the relationship between the infra sounds and the diagnosis of medical conditions. Obviously, it cannot be done theoretically. Hence, we can do it clinically. In order to show the clinical relationship between the heart sounds including infra sounds and the medical conditions, we need to check the relationship between those, which have been done in our study presented in the manuscript. It is, justifiably, not complete, but it shows the high probability. So, we conclude that more intensive studies are required. We have claimed almost identical claims in our previous responses to the comments from Reviewer 1.

Even though the complete relationship is not established in our study, we have shown high possibility of the close relationship. The complete one would be the final goal of the researchers in this research field. Every researcher would do something as one step to the final goal and we did one. Therefore, we believe, it is not appropriate to claim that we need to establish, theoretically or clinically, whether infrasound components are directly related to the diagnosis of medical conditions. This point was claimed in our answers to the previous responses for the first revision. 

2) Reviewer’s comment: Also the response to the other comments were unsatisfactory.

Author’s response: 

We are sorry that Reviewer 1 is not happy with our responses to his/her comments. However, we don’t see any scientific reason of his/her complaints. Therefore, we are receiving this sentence as an expression of his/her personal, emotional feeling, which we do not need to soothe.

---

## [Decision Letter · Decision Letter 2]

10 Apr 2024

PONE-D-23-23289R2Effects of precise cardio sounds on the success rate of phonocardiographyPLOS ONE

Dear Dr. Moon,

Thank you for submitting your manuscript to PLOS ONE. After careful consideration, we feel that it has merit but does not fully meet PLOS ONE’s publication criteria as it currently stands. Therefore, we invite you to submit a revised version of the manuscript that addresses the points raised during the review process.

We look forward to receiving your revised manuscript.

Kind regards,

Ali Mohammad Alqudah

Academic Editor

PLOS ONE

Journal Requirements:

Reviewers' comments:

Reviewer's Responses to Questions

**Comments to the Author**

1. If the authors have adequately addressed your comments raised in a previous round of review and you feel that this manuscript is now acceptable for publication, you may indicate that here to bypass the “Comments to the Author” section, enter your conflict of interest statement in the “Confidential to Editor” section, and submit your "Accept" recommendation.

Reviewer #2: All comments have been addressed

Reviewer #3: All comments have been addressed

2. Is the manuscript technically sound, and do the data support the conclusions?

Reviewer #2: Yes

Reviewer #3: Yes

3. Has the statistical analysis been performed appropriately and rigorously? 

Reviewer #2: Yes

Reviewer #3: Yes

4. Have the authors made all data underlying the findings in their manuscript fully available?

Reviewer #2: Yes

Reviewer #3: Yes

5. Is the manuscript presented in an intelligible fashion and written in standard English?

Reviewer #2: Yes

Reviewer #3: Yes

6. Review Comments to the Author

Reviewer #2: The paper is organized well, and it discussed the topic in different points. I recommend to accept it .

Reviewer #3: The authors have done good work on the title “Effects of precise cardio sounds on the success rate of phonocardiography”. It will add new knowledge and new areas of research to the subject area compared with other published material.

However, i have some minor concerns:

1.It would be more appropriate for the authors to define abbreviations upon first appearance in the main text such as micro-electromechanical systems (MEMS).

2.The authors have inserted Eq. 2 and 3 in the manuscript without citing them properly inside the manuscript. Kindly check it and perform the required amendment.

3.It would be more appropriate for the authors to define abbreviations of MATLAB as a proprietary multi-paradigm programming language and numeric computing environment developed by MathWorks. This will contribute more knowledge to the reader.

4.A minor inconsistency in the usage of the term "CWT” was noticed. In some sentences, it was written in capital letters, while in others, it was written in lowercase. To maintain consistency and clarity throughout the text, I suggest standardizing the writing of "cwt" by either capitalizing it or using lowercase letters consistently.

5.Moderate English grammar editing is required throughout the manuscript, for example:

a.In the section of train results, “However, for Sensitivity, there was a decrease in …). moderate editing is required.

Best regards,

7. PLOS authors have the option to publish the peer review history of their article (what does this mean?). If published, this will include your full peer review and any attached files.

Reviewer #2: No

Reviewer #3: **Yes: **MAI ABDEL HALEEM ABUSALAH

---

## [Author Response · Author response to Decision Letter 2]

16 May 2024

To Editor:

 Upon reviewing the references, it was identified that the previously cited references 5 and 9 were the same. Consequently, reference 9 has been amended to reference 5 in the revised document.

Revised manuscript: 

Heart sounds provided in PhysioNet can be successfully classified into two categories (normal and abnormal) with a 98.3% success rate by using long short-term memory (LSTM) and segmentation based on logistic-regression hidden semi-Markov model (HSMM)-based techniques. [5]

Reviewer #3:

<Review Comments>

1) Reviewer’s comment: It would be more appropriate for the authors to define abbreviations upon first appearance in the main text such as micro-electromechanical systems (MEMS).

Author’s response: 

We appreciate your attention to detail regarding the use of abbreviations in our manuscript. Upon your suggestion, we have revised the text to include a definition for MEMS at its first occurrence as follows: MEMS (Micro-Electromechanical Systems). Additionally, we conducted a thorough review of the document to ensure that all other necessary abbreviations are similarly defined at their first mention, thereby improving the readability and accessibility of our paper.

Revised manuscript: 

This design allows compatibility with the ½-inch B&K 4193 microphone and also with miniature MEMS(Micro-Electromechanical Systems) microphones and large microphones up to 1 inch in size.(Line Number 91 – 93)

2) Reviewer’s comment: The authors have inserted Eq. 2 and 3 in the manuscript without citing them properly inside the manuscript. Kindly check it and perform the required amendment.

Author’s response: 

We appreciate your observation regarding the lack of proper citation for Equations 2 and 3. To address this issue, we have revised the relevant sections to include and properly cite these equations, particularly in our explanation of the Continuous Wavelet Transform (CWT). This amendment ensures that all mathematical expressions are accurately referenced within the text, enhancing both the credibility and scholarly integrity of our manuscript. Thank you for bringing this to our attention.

Revised manuscript: 

Therefore, we used the continuous wavelet transform (CWT), which is defined as the integral (Eq. 2) over all time of the product of the signal f(t) and a scaled and translated version of the mother wavelet ψ. Here, ψb,a(t) represents the mother wavelet (Eq. 3) modified by the scale parameter a and translation parameter b, where a influences the dilation of the wavelet, allowing it to capture features at various frequencies, and b adjusts the wavelet’s position, facilitating the analysis of the signal’s temporal characteristics. [20] (Line Number 130 – 136)

3) Reviewer’s comment: It would be more appropriate for the authors to define abbreviations of MATLAB as a proprietary multi-paradigm programming language and numeric computing environment developed by MathWorks. This will contribute more knowledge to the reader.

Author’s response: 

We appreciate your suggestion to provide a more detailed explanation of MATLAB in our manuscript. In response to your feedback, we have included a comprehensive definition of MATLAB as follows: MATLAB, a proprietary multi-paradigm programming language and numerical computing environment, is developed and distributed by MathWorks. This enhancement aims to enrich the reader's understanding of the context in which MATLAB is utilized within our study. 

Revised manuscript: 

In the Wavelet Toolbox of MATLAB, a proprietary multi-paradigm programming language and numerical computing environment developed and distributed by MathWorks, the CWT function is utilized to perform the wavelet transform. (Line Number 146 – 148)

4) Reviewer’s comment: A minor inconsistency in the usage of the term "CWT” was noticed. In some sentences, it was written in capital letters, while in others, it was written in lowercase. To maintain consistency and clarity throughout the text, I suggest standardizing the writing of "cwt" by either capitalizing it or using lowercase letters consistently.

Author’s response: 

Thank you for pointing out the inconsistency in the use of the abbreviation "CWT." To address this, we have standardized the term to "CWT" throughout the manuscript when referring to the concept of Continuous Wavelet Transform. Additionally, regarding the term "dense" as it relates to TensorFlow's layers, we have corrected all instances to use lowercase "dense" consistently throughout the document. This distinction clarifies the text and adheres to the proper usage of technical terms. Your feedback has significantly improved the manuscript’s consistency and clarity. 

Revised manuscript: 

1. Therefore, we used the continuous wavelet transform (CWT), which is an algorithm that transforms a signal in the time domain to the time–frequency domain by translating and scaling a basis function called a wavelet. (Line Number 130 – 132)

2. In contrast, CWT allows for high-frequency resolution at low frequencies by multi-resolution , preventing distortion in the transformation results of the low-frequency range. (Line Number 143 – 145)

3. In the Wavelet Toolbox of MATLAB, a proprietary multi-paradigm programming language and numerical computing environment developed and distributed by MathWorks, the CWT function is utilized to perform the wavelet transform. The CWT function was implemented in a frequency range of 2–300 Hz, using a Morlet wavelet as the basis wavelet. (Line Number 148 – 150)

4. STFT and CWT for the heart-sound signal were obtained from the experimental equipment. Compared to the STFT image (Fig. 3A), the CWT image (Fig. 3B) provides a clearer representation of the frequency characteristics of the signal and enables feature extraction in both low- and high-frequency bands. (Line Number 153 – 156)

5. Different preprocessing techniques or models, such as those other than the currently-used CWT, CNN-dense, or CNN-LSTM, may achieve higher accuracy than those that do use them. (Line Number 333 – 335)

5) Reviewer’s comment: Moderate English grammar editing is required throughout the manuscript, for example: a.In the section of train results, “However, for Sensitivity, there was a decrease in …). moderate editing is required.

Author’s response: 

 We acknowledge your concerns regarding the grammatical accuracy in our manuscript, particularly in the sections added during the previous revision. We have taken your feedback seriously and have resubmitted the document to a professional editing service for thorough proofreading and correction. This ensures that the text now meets the high standard of clarity and grammaticality expected in scholarly communication.

---

## [Editor Report · Decision Letter 3]

30 May 2024

Effects of precise cardio sounds on the success rate of phonocardiography

PONE-D-23-23289R3

Dear Dr. Moon,

We’re pleased to inform you that your manuscript has been judged scientifically suitable for publication and will be formally accepted for publication once it meets all outstanding technical requirements.

Kind regards,

Ali Mohammad Alqudah

Academic Editor

PLOS ONE

---

## [Editor Report · Acceptance letter]

4 Jun 2024

PONE-D-23-23289R3 

PLOS ONE

Dear Dr. Moon, 

I'm pleased to inform you that your manuscript has been deemed suitable for publication in PLOS ONE. Congratulations! Your manuscript is now being handed over to our production team.

Kind regards, 

on behalf of

Dr. Ali Mohammad Alqudah 

Academic Editor

PLOS ONE